# Protective Role of p66Shc Deletion in Physiological Renal Aging: Effects on G Protein-Coupled Receptor 124 Expression and Associated Cellular Senescence

**DOI:** 10.3390/ijms262211096

**Published:** 2025-11-17

**Authors:** Carla Iacobini, Martina Vitale, Federica Sentinelli, Antonietta Lucarelli, Jonida Haxhi, Ilaria Sergio, Giuseppe Pugliese, Stefano Menini

**Affiliations:** 1Department of Clinical and Molecular Medicine, “La Sapienza” University, 00189 Rome, Italy; carla.iacobini@uniroma1.it (C.I.); martina.vitale@uniroma1.it (M.V.); antoniettalucarelli99@gmail.com (A.L.); jonida.haxhi@uniroma1.it (J.H.); 2Department of Clinical Medicine, Public Health, Life and Environmental Sciences (MeSVA), University of L’Aquila, 67100 L’Aquila, Italy; federica.sentinelli@univaq.it; 3Tumor Immunology and Immunotherapy Unit, IRCCS Regina Elena National Cancer Institute, 00144 Rome, Italy; ilaria.sergio@ifo.it

**Keywords:** cellular senescence, fibrosis, glomerulosclerosis, GPR124, healthspan, mitochondrial signaling, oxidative stress, podocytes

## Abstract

The adaptor protein p66Shc regulates oxidative stress and aging, but its role in renal aging is unclear. We investigated the effects of p66Shc deletion on age-related kidney changes, focusing on G protein-coupled receptor 124 (GPR124) and cellular senescence. Using kidney and urine samples from p66Shc^−/−^ and wild-type mice aged 5–24 months, we found that p66Shc deficiency slows renal aging. Compared to wild-type mice, p66Shc^−/−^ mice showed better kidney function and reduced glomerular sclerosis and mesangial expansion from 18 months, alongside attenuated fibrosis, oxidative stress, and podocyte loss. Age-related declines in GPR124 were also smaller in p66Shc^−/−^ mice, correlating with lower p16^INK4a^ levels in glomeruli and cultured podocytes. These findings suggest that p66Shc deletion provides kidney protection by limiting oxidative stress and senescence, potentially through the preservation of GPR124. This study links p66Shc to natural kidney aging and identifies GPR124 as a mediator of p66Shc-driven senescence, suggesting potential targets for interventions in age-related renal decline.

## 1. Introduction

The p66Shc protein is a mitochondrial adaptor that regulates oxidative stress responses and apoptosis [1]. Its deletion has been associated with reduced production of reactive oxygen species (ROS) and enhanced resistance to oxidative stress [2], both of which are key factors implicated in aging and age-related diseases [3]. Consistently, p66Shc deficiency has been shown to protect against age-related endothelial dysfunction [4,5], suggesting a broader role in mitigating oxidative stress-driven aging processes.

p66Shc was initially proposed as a gerontogene based on reports that p66Shc knockout (p66Shc^−/−^) mice exhibited extended lifespans compared to wild-type (WT) controls [6]. However, this interpretation has been challenged by subsequent studies that found no significant differences in median or maximum lifespan between the two genotypes, suggesting a more nuanced role for p66Shc in aging [7]. Supporting this, tissue-specific effects of p66Shc deletion have been documented [8], which may offset one another, resulting in a neutral net effect on overall lifespan [7]. Nonetheless, evidence from this and other studies indicates that the absence of p66Shc can influence healthspan, modulate age-related physiological changes, and affect the onset and progression of certain age-associated pathologies, including renal disease [7,9].

Regarding the biological functions of this gerontogene in renal pathophysiology, most existing research on p66Shc has focused on kidney disorders associated with diabetes, hypertension, and exposure to advanced glycation end products (AGEs) [10,11,12,13,14], all of which are known to accelerate renal aging and dysfunction. For instance, p66Shc ablation has been shown to protect renal mesenchymal stem cells from high glucose-induced oxidative stress, apoptosis, and senescence [11]. Furthermore, histological features of renal aging, such as mesangial expansion and glomerulosclerosis, were delayed or prevented in models of diabetic nephropathy [12]. Additionally, p66Shc^−/−^ mice exhibited protection against AGE-induced glomerulopathy by reducing oxidative tissue injury and limiting further AGE accumulation [13]. Although these models of accelerated aging imply a potential role for p66Shc in the aging process, studies directly examining the effects of p66Shc ablation on the natural, age-related decline in renal structure and function, independent of age-associated diseases, are lacking.

Recent evidence indicates that hyperglycemia-induced renal injury is mediated, at least in part, by the acceleration of glomerular senescence through mechanisms involving G protein-coupled receptor 124 (GPR124) [15]. Although GPR124 was not previously identified as a receptor associated with cellular senescence, this study suggests it plays a role in regulating podocyte senescence in the context of diabetic nephropathy. These findings position GPR124 as a promising therapeutic target for preventing renal aging and injury in diabetic nephropathy. Nevertheless, the impact of p66Shc ablation, a well-known gerontogene and modulator of cellular senescence signaling [11], on GPR124 expression and other senescence-related markers remains to be determined.

The primary aim of this study is to investigate the impact of p66Shc ablation on physiological renal aging, independent of confounding factors associated with disease models. As a secondary objective, the study explores the relationship between age-related structural and functional changes in the kidney and alterations in GPR124 expression. Furthermore, it assesses how p66Shc ablation influences these changes, including the appearance of senescence markers in glomeruli and cultured podocytes, which has not yet been elucidated.

## 2. Results

Before proceeding with the main analyses, all renal specimens underwent an initial screening to exclude those displaying signs of renal pathology, as such conditions could confound the interpretation of p66Shc’s role in age-related structural and functional alterations of the kidney, independently of unrelated disease processes. Based on standard histological evaluation, kidneys from seven p66Shc^−/−^ mice—three at 8 months (out of ten), two at 18 months (out of seven), and two at 24 months (out of nine)—as well as one WT mouse at 18 months (out of six), were excluded from subsequent analyses due to the presence of perivascular inflammatory infiltrates, indicative of ongoing infection or active immune/inflammatory responses (Appendix A).

### 2.1. Renal Function and Structure

Both genotypes exhibited age-related increases in urinary protein-to-creatinine and albumin-to-creatinine ratios. However, at 18 and 24 months, these levels were notably lower in p66Shc^−/−^ mice compared to age-matched WT mice, by 22% and 19% at 18 months, and by 21% and 26% at 24 months, respectively (Figure 1A).

Aging was associated with a marked increase in all structural parameters across both genotypes. Although the pattern of age-related changes in kidney structure was similar between the groups, the magnitude of these changes was markedly lower in p66Shc^−/−^ mice compared to WT controls. Consequently, significant renal lesions, such as PAS-positive deposits in the mesangium and glomerular sclerosis, particularly at the vascular pole, were evident in WT mice at 18 and, especially, 24 months of age, but were markedly reduced in p66Shc^−/−^ mice (Figure 1B). Specifically, mean glomerular area (mGA), mean mesangial area (mMA), fractional mesangial area (fMA), and the glomerulosclerosis index (GSI) were significantly reduced in p66Shc^−/−^ mice compared to age-matched WT controls, with reductions of 9%, 20%, 12%, and 24%, respectively, at 18 months. These differences were even more pronounced at 24 months, reaching 15%, 29%, 16%, and 31%, respectively (Figure 1C).

### 2.2. Renal Fibrosis, Podocyte Number, and Oxidative Stress Markers

The mRNA expression of collagen IV α1-chain (*Col4a1*), a marker of basement membrane-associated fibrosis, and transforming growth factor-β1 (*Tgfb1*), a key regulator of fibrotic remodeling, increased significantly with age in both genotypes. However, these age-related elevations were more pronounced in WT mice (43–44% increase at 24 vs. 5 months) compared to p66Shc^−/−^ mice (25–29% increase). Consequently, transcript levels were significantly lower in p66Shc^−/−^ mice than in WT counterparts beginning at 18 months of age (*Col4a1*: −10%; *Tgfb1*: −12%), with these differences persisting and slightly increasing by 24 months (*Col4a1*: −11%; *Tgfb1*: −13%) (Figure 2A).

Consistent with the mRNA data, collagen IV (COL-IV) protein levels in the glomeruli were significantly higher in WT mice compared to p66Shc^−/−^ mice at 18 months of age, whereas no difference was observed at 8 months (Figure 2B). In parallel with the increase in renal and glomerular fibrosis markers, the number of podocytes was significantly reduced by 21% in WT mice at 18 months, compared to a non-significant 5% reduction in p66Shc^−/−^ mice, resulting in a significant difference between the two genotypes (Figure 2C). Furthermore, glomerular levels of NADPH oxidase 4 (NOX4), a major source of ROS in renal cells [16], as well as the protein oxidation marker nitrotyrosine, were significantly lower in p66Shc^−/−^ mice relative to WT at 18 months, decreased by 51% and 62%, respectively (Figure 3A,B).

### 2.3. Renal Expression Levels of GPR124 and p16^INK4a^

GPR124 levels gradually declined with age in mouse kidney samples; however, this reduction was less pronounced in p66Shc^−/−^ mice compared to WT controls. Consequently, a significant difference in GPR124 expression between the two genotypes emerged starting at 18 months of age, with p66Shc^−/−^ mice exhibiting 36% and 39% higher receptor levels than WT mice at 18 and 24 months, respectively (Figure 4A).

Immunohistochemistry (IHC) analysis revealed that the glomerular expression of GPR124 significantly declines by approximately 60% between 8 and 18 months of age in WT mice. In contrast, p66Shc^−/−^ mice exhibited a more preserved GPR124 expression at 18 months, with only a 26% reduction over the same period (Figure 4B). Given the established association between GPR124 and cellular senescence [15], we examined the protein levels of p16^INK4a^, a well-characterized marker of senescence in renal cells. Our results reveal a progressive, age-dependent increase in p16^INK4a^ expression in the kidneys of both genotypes; however, this increase occurred more slowly in p66Shc^−/−^ mice. Notably, p66Shc deficiency resulted in a statistically significant reduction in p16^INK4a^ protein levels, with decreases of 26% at 18 months and 22% at 24 months compared to WT controls (Figure 4C). Consistently, the number of p16^INK4a^-positive glomerular cells, assessed by IHC, increased less in p66Shc^−/−^ mice (101%) than in WT mice (146%) between 8 and 18 months of age (Figure 4D).

### 2.4. Effect of p66Shc Silencing on the Expression Levels of GPR124 and Senescence-Related Markers in Cultured Podocytes

To explore the involvement of p66Shc in regulating GPR124 expression during cellular senescence, we induced premature senescence in primary human podocytes (PODO/TERT256 cell line) by treating them with Adriamycin (ADR), a DNA-damaging anthracycline [17]. To confirm activation of the senescence pathway in our model, we focused our analysis on key senescence-associated human genes: *CDKN2A*, which encodes the p16^INK4a^ protein; *CDKN2D*, which encodes p19^INK4d^; and *CDKN1A*, which encodes p21. In podocytes treated with non-targeting control siRNA (si-*NC*), ADR exposure resulted in a 55% increase in *CDKN2A* and a 39% increase in *CDKN1A* expression compared to untreated control (Ctr) cells, while *CDKN2D* levels remained largely unchanged. In contrast, p66Shc-silenced podocytes (si-*p66Shc*) showed only modest increases in *CDKN2A* (26%) and *CDKN1A* (12%) upon ADR treatment, with a statistically significant elevation observed only for *CDKN2A* relative to Ctr (Figure 5A). Senescence induction in podocytes was further confirmed by senescence-associated β-galactosidase (SA-β-gal) staining and the expression of p16^INK4a^. Notably, si-*p66Shc* substantially attenuated ADR-induced senescence, as evidenced by a reduction in SA-β-gal activity (−31%) (Figure 5B) and decreased expression of p16^INK4a^ by 27% compared with si-*NC* (Figure 5C). Interestingly, this reduction in senescence markers and activity in si-*p66Shc* podocytes following ADR exposure was paralleled by a corresponding increase in GPR124 protein expression (≈50%) compared to si-*NC*-treated cells (Figure 5D).

## 3. Discussion

Our study demonstrates that p66Shc ablation provides significant protection against physiological renal aging, as indicated by improved renal function, reduced structural damage, and decreased markers of fibrosis and oxidative stress in aging mice. Notable differences from WT mice in both structural and functional decline emerged at 18 months of age, accompanied by consistent reductions in fibrosis, oxidative stress markers, and a preservation of glomerular podocyte number. These findings extend previous reports that predominantly focused on accelerated renal aging models, such as diabetic nephropathy [12,13,18] and hypertension [19], by showing that p66Shc deficiency also slows natural, age-related renal decline independent of confounding disease states. This addresses a critical gap in the literature, as the role of p66Shc in intrinsic renal aging has not yet been investigated.

Consistent with its established role in mitochondrial ROS production and the regulation of oxidative stress [1,4,6,12], p66Shc^−/−^ mice exhibited significantly lower levels of NOX4 and nitrotyrosine at 18 months, a time point at which structural and functional parameters were significantly improved compared to WT mice. While earlier studies presented conflicting evidence regarding the impact of p66Shc on lifespan extension [6,7], our data support the view that p66Shc primarily influences healthspan and tissue-specific aging phenotypes, rather than lifespan itself [20]. The attenuated progression of glomerulosclerosis, mesangial expansion, and fibrosis marker expression (e.g., *Col4a1*, *Tgfb1*) in p66Shc^−/−^ mice highlights a direct role for this protein in modulating the structural remodeling associated with renal aging. However, consistent with previously reported adverse health effects and a predisposition to immune dysregulation affecting both the adaptive and innate immune compartments [21,22], we observed a higher incidence of perivascular inflammatory infiltrates, composed of monocytes and lymphocytes surrounding renal arteries, in p66Shc^−/−^ mice aged 8 months or older (7 out of 26), compared to age-matched WT controls (1 out of 21). These mice were excluded from the experimental group analyses to avoid confounding the interpretation of p66Shc’s role in age-related structural and functional kidney alterations (see Section 2 and Section 4). Nevertheless, this finding supports previous reports linking p66Shc deficiency to increased susceptibility to infections [22,23,24], as well as a heightened risk of immune-mediated inflammation, autoimmune diseases, and leukocyte proliferative disorders [21,25,26,27]. Moreover, this immune dysregulation, along with metabolic abnormalities reported in some studies [28], may help explain why the slower progression of age-related structural and functional decline observed in this and other investigations does not translate into extended lifespan in p66Shc mice under controlled laboratory conditions [7]. In naturalistic outdoor environments, these immune and metabolic vulnerabilities could even become detrimental, potentially contributing to negative selective pressure against this genotype [24].

Notably, we show for the first time that p66Shc deficiency mitigates the age-dependent decline of GPR124 expression in the kidney, paralleled by a slower increase in the senescence marker p16^INK4a^. These findings link p66Shc to the regulation of cellular senescence pathways, likely via preservation of GPR124 signaling, which has recently emerged as an important regulator of glomerular podocyte senescence [15,29]. Our in vitro experiments further support this mechanism: silencing p66Shc in cultured human podocytes attenuated ADR-induced senescence, as demonstrated by reduced mRNA levels of key senescence-associated genes, decreased SA-β-gal activity, lower p16^INK4a^ protein expression, and restored GPR124 expression. These findings indicate a direct role for p66Shc in regulating podocyte senescence pathways. Collectively, the data suggest that p66Shc promotes renal aging by enhancing oxidative stress and accelerating cellular senescence, potentially through the downregulation of GPR124. Furthermore, the results imply that p66Shc may function as a negative feedback regulator of GPR124 signaling. The preservation of GPR124 and attenuation of senescence markers in p66Shc-deficient mice indicate that targeting this pathway may offer therapeutic benefits to delay renal aging and reduce susceptibility to age-associated kidney diseases.

Our study has some limitations, including the absence of lifespan analysis. Notably, it does not clarify the precise mechanism by which p66Shc contributes to the loss of GPR124 in podocytes, whether through a direct interaction between the adaptor protein and GPR124 or indirectly via its role in modulating oxidative stress. Future studies should aim to elucidate the specific signaling pathways linking p66Shc, oxidative stress, GPR124, and cellular senescence. Nevertheless, our findings substantially advance the understanding of p66Shc’s role in renal aging and highlight the potential of p66Shc-related signaling pathways as modulators of renal aging.

## 4. Materials and Methods

### 4.1. Experimental Design

#### 4.1.1. In Vivo Study

Kidney tissue samples analyzed in this study were sourced from our biobank of formalin-fixed, paraffin-embedded (FFPE) specimens. These samples, along with corresponding urine specimens, were originally collected from our colony of p66Shc^−/−^ and SV/129 WT mice as part of previously published studies by our group, which explored the effects of p66Shc gene deletion on renal pathophysiology [12,13]. All specimens were stored under standardized conditions to preserve sample integrity and ensure reproducibility of downstream analyses. In this study, urine samples (*n* = 5–8 per group) and kidney tissue samples (*n* = 5–11 per group) from p66Shc^−/−^ and wild-type male mice aged 5, 8, 18, and 24 months were analyzed. After excluding mice with renal specimens displaying signs of perivascular inflammation (see Section 2 and Appendix A), quantitative analyses of renal function and structure were conducted on the remaining samples (*n* = 5–6 per group for functional assessments; *n* = 5–10 per group for structural analyses). These sample sizes are consistent with established standards in nephropathy models and provide adequate statistical power to detect significant differences in key structural and functional parameters [30]. Molecular biology analyses, including assessments of gene expression and protein levels, were conducted on five renal samples per group. IHC was performed on kidneys from five mice per genotype at 8 and 18 months of age to evaluate the expression of markers associated with oxidative stress, fibrosis, senescence, podocytes, and GPR124. These time points correspond to the interval (months 8–18) during which structural and functional differences in the kidneys between the two genotypes reached statistical significance.

#### 4.1.2. In Vitro Study

The human podocyte cell line PODO/TERT256 (Evercyte GmbH, Vienna, Austria; CHT-033-0256) was used in this study. These cells, derived from human kidneys, exhibit stable proliferative capacity without losing key physiological features of primary podocytes, including the sustained expression of characteristic markers and functions. Cells were maintained at 37 °C in a humidified atmosphere containing 5% CO_2_ and 95% air, using collagen I-coated 100 mm^2^ culture dishes. The culture medium consisted of PodoUp3 (Evercyte GmbH, Leberstraße 20, 1110 Vienna, Austria, # MHT-033-3), supplemented with 10% fetal bovine serum, 100 U/mL penicillin, and 100 μg/mL streptomycin. Mycoplasma contamination was monitored biweekly via Real-Time PCR using the MycoSPY Kit (Biontex, Munich, Germany). Silencing of p66Shc (si-*p66Shc*) was achieved using Silencer^®^ Select Validated siRNA (Thermo Fisher Scientific Inc., Waltham, MA 02451, USA, Assay ID: s12811; # 4392420) and a non-targeting control siRNA (si-*NC* # 4390843) in antibiotic-free medium, using Lipofectamine^®^ RNAiMAX (Invitrogen, Carlsbad, CA, USA) as per manufacturer’s instructions. To induce premature senescence, podocytes were treated with 1 μg/mL ADR for 24 h [17]. Podocytes in the control group were maintained in standard growth media for the same period. Following treatment, the ADR-containing medium was replaced with fresh complete medium, which was renewed every three days over a subsequent six-day recovery period. Senescence markers and associated molecular targets were assessed on day 7.

### 4.2. Renal Function and Structure

Urine creatinine concentrations were determined using the Mouse Creatinine ELISA Kit (Abcam, Cambridge, UK, ab287790). Protein and albumin levels in urine were quantified using the Bradford dye-binding protein assay kit (Pierce, Rockford, IL, USA) and the Mouse Albumin ELISA Quantitation Kit #RK04196 (ABclonal Biotechnology, Dusseldorf, Germany), respectively. Values were normalized to urine creatinine levels. Kidney structural analysis was performed on multiple 4-μm tissue sections stained with periodic acid–Schiff (PAS) by an experienced pathologist who was blinded to group allocations. Glomerular sclerosis was evaluated using a semiquantitative approach based on the examination of 100 glomeruli per animal [31]. Each glomerulus was scored on a scale from 0 to 4, reflecting the extent of cross-sectional sclerosis: 0 (none), 1 (<25%), 2 (25–50%), 3 (51–75%), and 4 (>75%). A glomerular sclerosis index (GSI) was then calculated for each sample using the formula: (N1 × 1 + N2 × 2 + N3 × 3 + N4 × 4)/100, where N1 through N4 represent the number of glomeruli assigned each respective score. Glomerular area and mesangial expansion were evaluated using the Image Pro Premier 9.2 interactive image analysis system (Immagini&Computer, Milan, Italy) [32]. For each sample, at least 60 glomerular tuft profiles were analyzed, and the harmonic mean of their areas was calculated to determine the mean glomerular area (mGA). Periodic acid–Schiff (PAS)-positive regions within these glomeruli were quantified and expressed as a percentage of the total tuft area, referred to as the fractional mesangial area (fMA). A color threshold was established by selecting three to five representative pixels from PAS-positive regions. The mean mesangial area (mMA) was then derived using the formula: (fMA × mGA)/100 [30].

### 4.3. Immunostaining

The primary and secondary antibodies used for immunostaining and Western blotting are listed in Appendix A. IHC was performed on mouse kidney tissue sections to detect COL-IV, NOX4, nitrotyrosine, GPR124, and the senescence marker p16INK4a. Following heat-induced antigen retrieval, the sections were incubated overnight at 4 °C with specific primary antibodies. The next day, sections were incubated for 1 h at room temperature with appropriate biotinylated secondary antibodies. To ensure antibody specificity, control samples were prepared using non-immune serum in place of primary antibodies. Sections were analyzed under a Nikon Eclipse E600 light microscope (Nikon, Tokyo, Japan). Glomerular expression of the podocyte marker Wilms Tumor 1 (WT1) was evaluated by dual-label immunofluorescence (IF) [33]. To minimize tissue autofluorescence, slides were pre-treated with 0.1% Sudan Black B for 5–10 min in the dark before immunostaining. Imaging was performed using a Zeiss Axiovert 200 M fluorescence microscope fitted with a 25 x/N.A. 0.95 objective and an Axiocam 503 color camera, operated via ZEN 2.0 (blue edition) software (Zeiss, Milan, Italy). For the quantification of IHC staining, a method similar to that used for assessing PAS positivity was employed. Using the Image Pro Premier 9.2 analysis system, a region of interest was manually outlined around each glomerulus. The proportion of the glomerular area exhibiting positive staining was then determined at 400 X magnification based on a predefined color threshold [32]. For each kidney sample, the mean percentage of stained glomerular area was calculated from a minimum of 60 glomeruli. In the case of the p16INK4a and WT1, glomerular positive cells were counted at 400 X magnification and expressed as a percentage of total glomerular cells.

### 4.4. Western Blot

Proteins were extracted from FFPE tissue using the Qproteome FFPE Tissue Kit (Qiagen, Milan, Italy), in accordance with the protocol provided by the manufacturer. Western blotting was performed according to the protocols recommended by the antibody suppliers. Briefly, proteins extracted from FFPE kidney tissue and cultured podocytes were separated by SDS-PAGE and subsequently transferred onto PVDF membranes [30]. Membranes were blocked with 5% skim milk, and incubations were performed using either 5% skim milk or BSA, as recommended by the antibody and supplier. Primary antibodies for GPR124 and p16INK4a were applied overnight at 4 °C, followed by incubation with a biotinylated secondary antibody at room temperature for one hour. Signal detection was performed using Clarity or Clarity Max enhanced chemiluminescence substrates (Bio-Rad Laboratories, Milan, Italy). The resulting bands were visualized and quantified using the ChemiDoc XRS imaging system (Bio-Rad Laboratories, Hercules, CA 94547, USA).

### 4.5. SA-β-Gal Staining

Senescence-associated β-galactosidase (SA-β-gal) staining was performed using the Senescence Cells Histochemical Staining Kit (Cat. No. CS0030, Sigma-Aldrich, St. Louis, MO, USA), following the protocol provided. In summary, PODO/TERT256 cells cultured on slides were rinsed twice with PBS and fixed using the supplied fixative solution for 15 min at room temperature following the designated treatments. Subsequently, each slide was incubated overnight at 37 °C in a non-CO_2_ incubator with 1 mL of β-galactosidase staining solution. After incubation, slides were washed twice with PBS, and images were captured using a light microscope. For quantification, ten random fields per chamber were imaged and analyzed by blinded counting of SA-β-gal-positive cells.

### 4.6. RT-PCR

The mRNA expression levels of *Col4a1* and *Tgfb1* in renal tissue, along with those of the senescence-associated genes *CDKN2A* (p16INK4a), *CDKN2D* (p19INK4d), and *CDKN1A* (p21) in PODO/TERT256 cells, were assessed by RT-PCR using TaqMan Gene Expression Assays (Applied Biosystems, Carlsbad, CA, USA), listed in Appendix A. Total RNA from FFPE kidney tissue was extracted using the RNeasy FFPE Kit (Qiagen, Milan, Italy) according to the manufacturer’s protocol, while RNA from PODO/TERT256 cells was isolated using the RNeasy Plus Mini Kit (Qiagen, Milan, Italy). Reverse transcription of RNA to complementary DNA (cDNA) was performed with the High-Capacity cDNA Reverse Transcription Kit (Thermo Fisher Scientific). Quantitative real-time PCR was performed on a StepOne Real-Time PCR System (Thermo Fisher Scientific), and relative gene expression was calculated using the ΔΔCt method, with β-actin serving as the endogenous control.

### 4.7. Statistical Analysis

The number of mice and independent biological replicates used in each experiment is detailed in the figure legends. Data are presented as mean ± standard deviation (SD). Additionally, the percent change between p66Shc^−/−^ and WT animals was calculated. Comparisons between p66Shc^−/−^ and WT mice at each age were conducted using the Holm–Sidak method with α = 0.05. Each comparison was analyzed independently, without assuming equal variances. For analyses involving more than two groups (e.g., different age groups or cell treatments), one-way or two-way ANOVA was performed as appropriate, followed by Tukey’s multiple comparison test. A *p*-value of less than 0.05 was considered statistically significant. All statistical analyses were performed on raw datasets using GraphPad Prism version 8.00 for Windows (GraphPad Software, San Diego, CA, USA).

## 5. Conclusions

Our findings suggest that p66Shc promotes renal aging by enhancing oxidative stress and accelerating cellular senescence, potentially via downregulation of GPR124 (Figure 6).

The preservation of GPR124 expression and the reduction in senescence markers in p66Shc-deficient mice highlight the relevance of this pathway. Targeting p66Shc–GPR124 signaling may represent a promising therapeutic strategy for delaying renal aging and mitigating age-related kidney diseases.

## Figures and Tables

**Figure 1 ijms-26-11096-f001:**
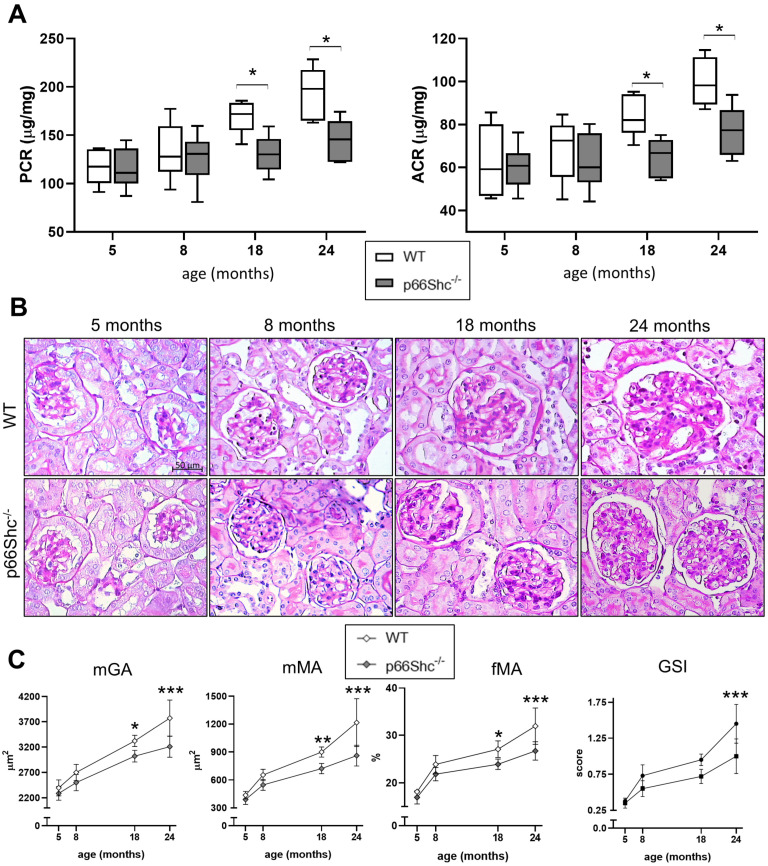
**Renal functional and structural changes in WT and p66Shc^−/−^ mice across aging**. Urinary protein/creatinine ratio (PCR, µg/mg) and albumin/creatinine ratio (ACR, µg/mg) (**A**). Representative images of periodic acid–Schiff (PAS) staining (**B**). Quantification of mean glomerular area (mGA, µm^2^), mean mesangial area (mMA, µm^2^), fractional mesangial area (fMA, %), and glomerulosclerosis index (GSI, score) in wild-type (WT) and p66Shc^−/−^ mice (**C**). Data were obtained from animals aged 5, 8, 18, and 24 months (*n* = 5–6 per group for functional analyses; *n* = 5–10 per group for structural analyses). In (**A**), box plots display the median (center line), interquartile range (box limits), and minimum and maximum values (whiskers). Data in (**C**) are presented as mean ± SD. * *p* < 0.05, ** *p* < 0.01, *** *p* < 0.001. Statistical comparisons between genotypes at each age were performed using the Holm–Sidak method (α = 0.05).

**Figure 2 ijms-26-11096-f002:**
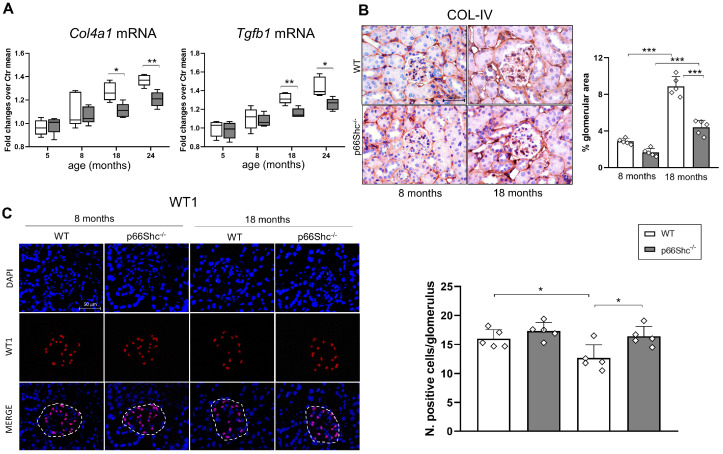
**Expression of *Col4a1* and *Tgfb1* mRNA, COL-IV protein, and WT1 in WT and p66Shc^−/−^ mice**. mRNA levels of *Col4a1* and *Tgfb1* measured between 5 and 24 months (**A**). Immunohistochemical staining and quantification of COL-IV protein levels at 8 and 18 months (**B**). Immunofluorescence detection and quantification of WT1 expression at 8 and 18 months (**C**). For all panels, *n* = 5 mice per group; in (**A**), box plots show the median (center line), interquartile range (box), and minimum/maximum values (whiskers). Data in (**B**,**C**) are presented as mean ± SD, and each dot represents one animal. Statistical comparisons in (**A**) were made between genotypes at each age using the Holm–Sidak method (α = 0.05). One-way ANOVA followed by Tukey’s post hoc test was performed for (**B**,**C**). * *p* < 0.05, ** *p* < 0.01 *** *p* < 0.001.

**Figure 3 ijms-26-11096-f003:**
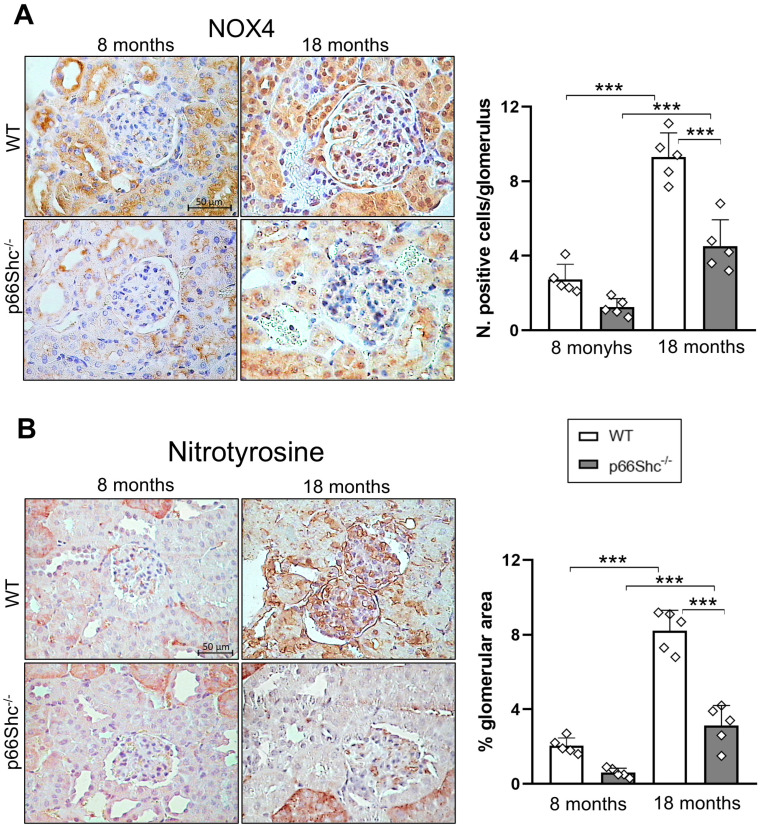
**Reduced renal glomerular NOX4 and nitrotyrosine levels in p66Shc^−/−^ mice**. Representative immunohistochemical staining for NOX4 (**A**) and nitrotyrosine (**B**) in kidney sections from WT and p66Shc^−/−^ mice. Quantification of staining intensity is shown for each marker. Data represent mean ± SD (*n* = 5 mice per group), with each dot corresponding to an individual animal. Statistical analysis was performed using one-way ANOVA followed by Tukey’s post hoc test. *** *p* < 0.001.

**Figure 4 ijms-26-11096-f004:**
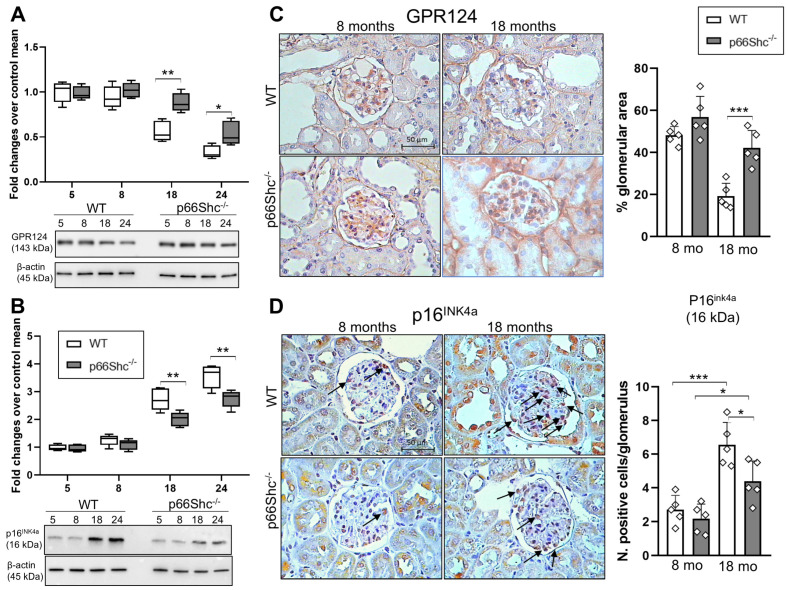
**Age-dependent changes in GPR124 and p16^Ink4a^ expression in WT and p66Shc^−/−^ mouse kidneys**. Representative Western blot images and quantification of GPR124 (**A**) and p16Ink4a (**B**) expression in kidney samples from wild-type (WT) and p66Shc^−/−^ mice at different ages. Representative immunohistochemical staining and quantification of GPR124 (**C**) and p16^Ink4a^ (**D**) in kidney sections from WT and p66Shc^−/−^ mice at 8 and 18 months of age. Arrows in (**D**) highlight immunopositive cells. For all panels, *n* = 5 mice per group. In (**A**) and (**B**), box plots display the median (center line), interquartile range (box), and minimum/maximum values (whiskers). Data in (**C**,**D**) are presented as mean ± SD, and each dot represents one animal. Statistical comparisons in (**A**,**B**) were performed between genotypes at each age using the Holm–Sidak method (α = 0.05). One-way ANOVA followed by Tukey’s post hoc test applied to (**C**,**D**). * *p* < 0.05, ** *p* < 0.01, *** *p* < 0.001.

**Figure 5 ijms-26-11096-f005:**
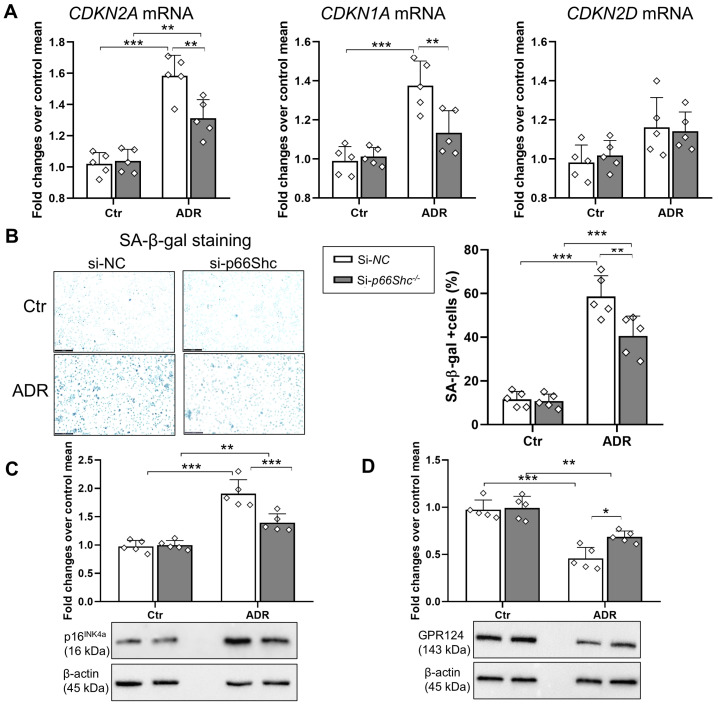
**Silencing of p66Shc modulates senescence markers and GPR124 expression in podocytes exposed to Adriamycin**. mRNA expression levels of *CDKN2A*, *CDKN1A*, and *CDKN2D* (**A**); SA-β-galactosidase (SA-β-gal) staining (**B**); representative Western blot images and quantification of GPR124 (**C**) and p16^Ink4a^ (**D**) protein expression in podocytes transfected with siRNA targeting p66Shc (si-*p66Shc*) or control siRNA (si-*NC*), and either left untreated (Ctr) or treated with Adriamycin (ADR, 1 μg/mL) for 24 h. Data are presented as mean ± SD (*n* = 5 biological replicates per group), with each dot representing an individual sample. Statistical analysis was performed using two-way ANOVA followed by Tukey’s post hoc test. * *p* < 0.05, ** *p* < 0.01, *** *p* < 0.001.

**Figure 6 ijms-26-11096-f006:**
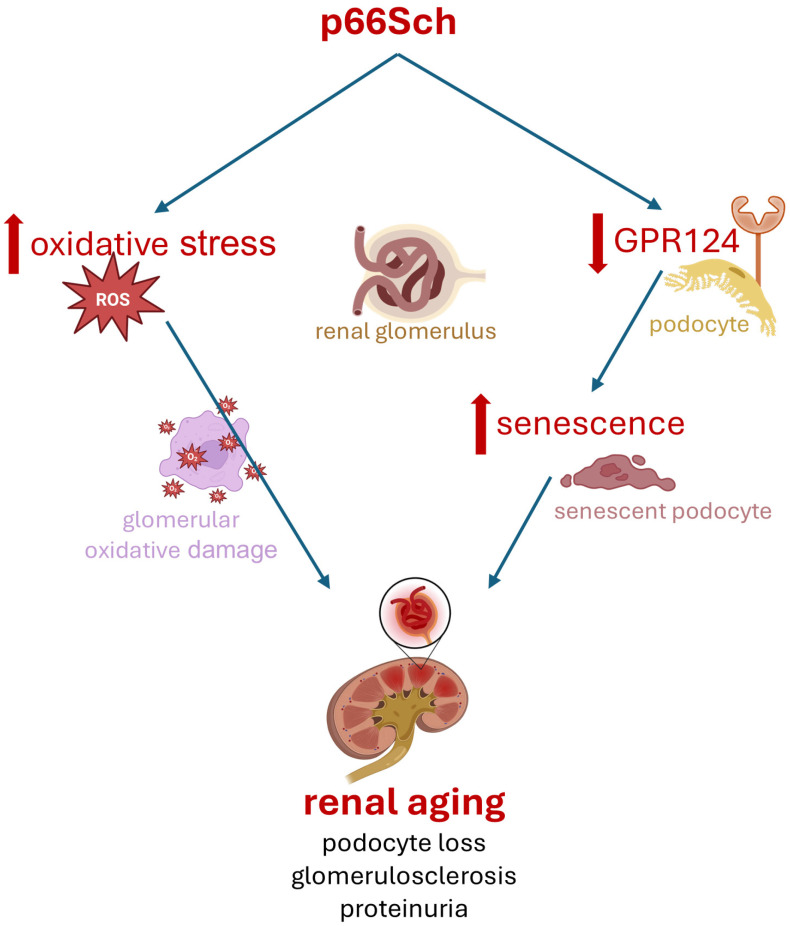
**Proposed mechanism by which p66Shc contributes to renal aging.** Created in BioRender. Frattale, A. (2025) https://BioRender.com/ntic3e7. Accessed on 10 November 2025.

## Data Availability

The original data used to support the findings of this study are available from the corresponding authors [GP and SM] upon reasonable request.

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
