# Peer review of "Protective Role of p66Shc Deletion in Physiological Renal Aging: Effects on G Protein-Coupled Receptor 124 Expression and Associated Cellular Senescence"

_ijms, 2025, doi:10.3390/ijms262211096_

Round 1
Reviewer 1 Report
Comments and Suggestions for Authors
Reviewer comments
The study explores the role of the adaptor protein p66Shc in renal aging and investigates how its deletion affects kidney function over time. Authors compared p66Shc⁻/⁻ mice with wild-type mice aged 5–24 months and found that p66Shc deficiency slowed age-related kidney damage, including reduced glomerular sclerosis, fibrosis, oxidative stress, and podocyte loss. They observed that p66Shc⁻/⁻ mice also showed smaller age-related declines in GPR124 expression and lower levels of the senescence marker p16INK4a. They suggest that maintaining GPR124 may help limit cellular senescence in the kidney. The study concludes that p66Shc deletion protects against renal aging by reducing oxidative stress and senescence.
Decision: Minor revision
- Please explain whether p66Shc directly influences GPR124 or if the effect happens indirectly through changes in oxidative stress?
- The authors note that GPR124 is linked to diabetic kidney disease, but it is not clear why this receptor was chosen to study normal kidney aging. Could the authors explain the reason for selecting GPR124 in the context of natural (non-disease) aging?
- Previous research has shown mixed results about whether deleting p66Shc affects how long animals live. Could the authors please clarify whether this study is focused on improving the quality of health during aging (health span) or on changing overall lifespan specifically in relation to the kidneys?
- The authors excluded several kidney samples due to inflammation. Could they clarify whether this exclusion affected group sizes or statistical power, especially in older mice?
- The results show strong protection in p66Shc−/− mice, but it’s not clear if sex differences were considered. Were both male and female mice used, and if so, were results analysed separately?
- Silencing p66Shc increased GPR124 in podocytes. Is this effect specific to podocytes, or might it also occur in other kidney cell types?
- Do the authors expect that p66Shc silencing could similarly reduce senescence in other kidney cell types, like mesangial or tubular cells, or is this effect specific to podocytes. Please explain
- A mechanistic figure is needed to conclude in the study for the common reader of your paper.
Author Response
We thank the Reviewer for her/his positive comments and useful suggestions.
1. Please explain whether p66Shc directly influences GPR124 or if the effect happens indirectly through changes in oxidative stress?
We agree with the Reviewer that this represents a limitation of our study, which we have now fully acknowledged in the Discussion section (see Page 9, lines 268–271 of the revised manuscript). Our current data do not provide insights into how p66Shc may influence GPR124 expression.
2. The authors note that GPR124 is linked to diabetic kidney disease, but it is not clear why this receptor was chosen to study normal kidney aging. Could the authors explain the reason for selecting GPR124 in the context of natural (non-disease) aging?
As noted in the Introduction, recent evidence indicates that hyperglycemia-induced renal injury is mediated, at least in part, by accelerated glomerular senescence through mechanisms involving G protein-coupled receptor 124 (GPR124). Given that p66Shc is a well-established gerontogene, we hypothesized that it may influence natural podocyte and kidney aging specifically by modulating the senescence-associated regulator GPR124. We have now made this intent more explicit in the introduction (see Page 2, lines 70–72 of the revised manuscript): “Nevertheless, the impact of p66Shc ablation, a well-known gerontogene and modulator of cellular senescence signaling, on GPR124 expression and other senescence-related markers remains to be determined.” On the other hand, diabetes is often used as a model of accelerated aging, as it anticipates the onset and accelerates the progression of numerous age-related diseases, such as atherosclerosis and kidney damage.
3. Previous research has shown mixed results about whether deleting p66Shc affects how long animals live. Could the authors please clarify whether this study is focused on improving the quality of health during aging (health span) or on changing overall lifespan specifically in relation to the kidneys?
We agree with the Reviewer’s concerns and appreciate her/his insightful comment. As discussed also in the original manuscript, another limitation of our study is the absence of a lifespan analysis (page 9, line 268 of the revised manuscript). Considering this limitation, together with the observation that p66Shc⁻/⁻ mice are more susceptible to infections and immune/inflammatory conditions, we believe that our findings support the hypothesis that “p66Shc primarily influences healthspan and tissue-specific aging phenotypes, rather than lifespan itself” (page 8, lines 229–231 of the revised manuscript).
4. The authors excluded several kidney samples due to inflammation. Could they clarify whether this exclusion affected group sizes or statistical power, especially in older mice?
Thank you for your comment. In fact, the majority of exclusions occurred mainly in the 8-month group. However, all groups still consisted of at least five animals (5–11), as indicated in the experimental design. In our experience, this number is sufficient to ensure adequate statistical power to detect significant differences in the key structural and functional parameters (Ref. 30 of the manuscript of the revised manuscript).
5. The results show strong protection in p66Shc−/− mice, but it’s not clear if sex differences were considered. Were both male and female mice used, and if so, were results analysed separately?
The mice were all males, as reported in the experimental design (see Page 9, lane 286 of the revised manuscript).
6. Silencing p66Shc increased GPR124 in podocytes. Is this effect specific to podocytes, or might it also occur in other kidney cell types?
To the best of our knowledge, no data have been reported in the literature on the effect of p66Shc ablation on GPR124 expression in renal cells, apart from our own findings.
7. Do the authors expect that p66Shc silencing could similarly reduce senescence in other kidney cell types, like mesangial or tubular cells, or is this effect specific to podocytes. Please explain
We are unable to address this question, as our analyses were limited to podocytes, which play a key role in in age‐related glomerular damage (PMID: 35166234) and diabetic nephropathy (PMID: 39828038). Furthermore, we cannot exclude a possible effect of p66Shc on GPR124 expression in other renal cell types or tissues.
8. A mechanistic figure is needed to conclude in the study for the common reader of your paper.
As requested, we have included a mechanistic figure (see Figure 6).
Reviewer 2 Report
Comments and Suggestions for Authors
This manuscript is well designed and provides evidence that pp6Shc deletion delays physiological renal aging by preserving GPR124 expression and reducing oxidative stress-induced cellular senescence, which is physiologically relevant.
The authors reported the increased inflammatory infiltration in p66Shc -/- mice, which appears paradoxical given the reduction of oxidative stress. Since oxidative stress and inflammation are typically interdependent and mutually reinforcing, it would be helpful if the authors could clarify this “anti-oxidative” yet “pro-inflammatory” phenotypes in kidneys in ppShc -/- mice.
Author Response
We thank the Reviewer for the positive comment to our work.
The authors reported the increased inflammatory infiltration in p66Shc -/- mice, which appears paradoxical given the reduction of oxidative stress. Since oxidative stress and inflammation are typically interdependent and mutually reinforcing, it would be helpful if the authors could clarify this “anti-oxidative” yet “pro-inflammatory” phenotypes in kidneys in p66Shc -/- mice.
Animals exhibiting perivascular inflammatory infiltration were excluded from all analyses (structural, functional, immunohistochemical, etc.), as detailed in the Results and Experimental Design sections, to avoid potential confounding effects of pathological conditions (e.g., increased susceptibility to infections or autoimmune disorders) to which p66-deficient mice are more prone (Refs. 21–27 of the revised manuscript). These conditions could otherwise influence the assessment of the natural, pathology-free aging process of the kidney (see Page 2, lines 81–89, and Page 9, lines 290–293 of the revised manuscript). Notably, the increased perivascular inflammation observed in the excluded mice may represent a direct consequence of reduced superoxide production in macrophages of p66Shc⁻/⁻ mice, which compromises their host defense capacity.